Multimodal image fusion for enhanced vehicle identification in intelligent transport

Mudawi Naif Al 1
Ahmed Muhammad Waqas 2 222633@students.au.edu.pk
http://orcid.org/0000-0002-6503-2826 Alhasson Haifa F. 3
Alshassari Naif S. 4
Alazeb Abdulwahab 1 afalazeb@nu.edu.sa
Alshehri Mohammed 1
Alabdullah Bayan 5
1 Department of Computer Science, College of Computer Science and Information System, Najran University , Najran , Saudi Arabia
2 Department of Computer Science, Air University , Islamabad , Pakistan
3 Department of Information Technology, College of Computer, Qassim University , Buraydah , Saudi Arabia
4 Department of Cyber Security, College of Humanities, Umm Al-Qura University , Mecca , Saudi Arabia
5 Department of Information Systems, College of Computer and Information Sciences, Princess Nourah bint Abdulrahman University , Riyadh , Saudi Arabia
Sergi Consolato
Electronic publication date: 2025 Oct 30
Publication date: 2025
Volume: 11
Electronic Location ID: e3270
Received 2025 Mar 28; Accepted 2025 Sep 15
Copyright: © 2025 Mudawi et al.
Copyright year: 2025
Copyright holder: Mudawi et al.
License: This is an open access article distributed under the terms of the Creative Commons Attribution License, which permits unrestricted use, distribution, reproduction and adaptation in any medium and for any purpose provided that it is properly attributed. For attribution, the original author(s), title, publication source (PeerJ Computer Science) and either DOI or URL of the article must be cited.
License URL: https://creativecommons.org/licenses/by/4.0/

Keywords: Deep learning, Unmanned aerial vehicles, Remote sensing, Dynamic environments, Path planning, Multi-objects recognition

Funding: Princess Nourah bint Abdulrahman University, Riyadh, Saudi Arabia PNURSP2025R440 Najran University NU/GP/SERC/13/18-6 Princess Nourah bint Abdulrahman University Researchers Supporting Project number (PNURSP2025R440), Princess Nourah bint Abdulrahman University, Riyadh, Saudi Arabia. The APC was funded by the Deanship of Graduate Studies and Scientific Research at Najran University through the Nama’a program, with the project code NU/GP/SERC/13/18-6. The funders had no role in study design, data collection and analysis, decision to publish, or preparation of the manuscript.

==============================
Target detection in remote sensing is essential for applications such as law enforcement, military surveillance, and search-and-rescue. With advancements in computational power, deep learning methods have excelled in processing unimodal aerial imagery. The availability of diverse imaging modalities including, infrared, hyperspectral, multispectral, synthetic aperture radar, and Light Detection and Ranging (LiDAR) allows researchers to leverage complementary data sources. Integrating these multi-modal datasets has significantly enhanced detection performance, making these technologies more effective in real-world scenarios. In this work, we propose a novel approach that employs a deep learning-based attention mechanism to generate depth maps from aerial images. These depth maps are fused with RGB images to achieve enhanced feature representation. For image segmentation, we use Markov Random Fields (MRF), and for object detection, we adopt the You Only Look Once (YOLOv4) framework. Furthermore, we introduce a hybrid feature extraction technique that combines Histogram of Oriented Gradients (HOG) and Binary Robust Invariant Scalable Keypoints (BRISK) descriptors within the Vision Transformer (ViT) framework. Finally, a Residual Network with 18 layers (ResNet-18) is used for classification. Our model is evaluated on three benchmark datasets Roundabout Aerial, AU-Air, and Vehicle Aerial Imagery Dataset (VAID) achieving precision scores of 98.4%, 96.2%, and 97.4%, respectively, for object detection. Experimental results demonstrate that our approach outperforms existing state-of-the-art methods in vehicle detection and classification for aerial imagery.

Introduction

Detecting vehicles in aerial images is a crucial yet complex task due to the small size of targets, varying object scales, and intricate backgrounds in large-scale scenes. Traditional RGB images, commonly used in aerial imagery, capture visual details but often struggle in low-light conditions or when vehicles blend into the background. These limitations make it challenging to reliably detect and classify vehicles in environments with poor contrast or complex lighting conditions (Garrote et al., 2018; Ahmed & Jalal, 2024a).

To address these challenges, depth maps have emerged as a complementary modality, providing spatial information by encoding the distance between objects and the imaging sensor. Aerial platforms such as drones, satellites, and aircraft are equipped with advanced sensors that generate both RGB images and depth maps, enhancing the context provided by visual imagery (Roy et al., 2022). However, depth maps also have limitations they can suffer from noise, occlusions, and insufficient detail when distinguishing between objects and background in high-illumination and texture limited conditions (Qian et al., 2021).

The integration of RGB images and depth maps through multi-modal fusion addresses these shortcomings by combining the strengths of both modalities. Fusing RGB data, which provides rich color and texture, with depth information, which delivers spatial awareness, results in a more comprehensive dataset for vehicle detection (Person et al., 2019; Cao et al., 2019). This fusion is especially critical in aerial imagery, where vehicles can be obscured by urban infrastructure, vegetation, or uneven terrain. By leveraging both types of data, multi-modal fusion enables more accurate differentiation between vehicles and their surroundings, improving detection in environments that pose visual or spatial challenges (Ahmed et al., 2024; Setiawan, 2020).

In practical implementations such as Intelligent Transportation System (Dimitrakopoulos & Demestichas, 2010), multi-modal fusion has been an invaluable tool. Automated and timely identification of vehicles from aerial images is useful for traffic surveillance, traffic control, and enforcement of law and order. Aerial images fused with depth maps provide a more complete understanding of the scene, enabling better vehicle tracking and infrastructure assessment. This approach proves most effective in scenarios where there exists a dense infrastructural setup or occlusions obstructing the detector’s view, as well as rural settings characterized by fluctuating lighting conditions and terrain undulations (Al Mudawi et al., 2024; Fan et al., 2020).

Despite recent progress in aerial object detection and classification, three key gaps remain: The lack of monocular depth estimation approaches explicitly adapted to aerial viewpoints, which limits geometric understanding in the absence of stereo or Light Detection and Ranging (LiDAR) data.

Insufficient fusion strategies capable of preserving fine boundary detail when combining RGB and depth modalities, which is critical for detecting small and partially occluded vehicles.

Limited use of feature extraction frameworks that combine global contextual reasoning with explicit local shape descriptors, reducing discriminability for visually similar small-object classes.

This work addresses these gaps through an attention-based monocular depth generation module tailored for aerial scenes, an edge-preserving guided fusion mechanism, and a hybrid Vision Transformer embedding augmented with Histogram of Oriented Gradients (HOG) and Binary Robust Invariant Scalable Keypoints (BRISK) descriptors. The resulting pipeline delivers both geometric robustness and fine-grained classification capability for real-world intelligent transportation applications.

While our approach effectively tackles these specific issues, multi-modal fusion as a broader field is not without its challenges. The fusion process must contend with environmental factors such as changing lighting conditions, shadows, and occlusions, all of which can degrade both RGB and depth image quality. Moreover, the computational demands of fusing large-scale aerial imagery and depth data especially for real-time applications present significant hurdles. Yet, recent advances in machine learning, particularly the use of deep learning models like Convolutional Neural Networks (CNNs), have made it possible to efficiently process and fuse these datasets, significantly improving detection performance. To overcome these obstacles, we propose a novel deep learning-based architecture for robust and accurate vehicle detection and classification. The key contribution of our article are as follows. We propose a novel approach to generating depth maps from aerial images using a deep learning-based architecture. Our model is designed to estimate the depth of each pixel in an aerial image by utilizing a combination of convolutional layers and attention mechanisms.

We propose a novel feature extraction approach that integrates Histogram of Oriented Gradients (HOG) and Binary Robust Invariant Scalable Key points (BRISK) within the patch embedding layer of the Vision Transformer (ViT).

A comprehensive vehicle detection and classification pipeline that combines advanced preprocessing techniques, multi-modal fusion, segmentation, and deep learning-based detection and classification

The article is structured as follows: consequently, ‘Related Work’ of this article provides a review of the literature on target detection. ‘Materials and Methods’ introduces the selected methodology in detail with a brief evaluation of its advantages and limitations. In ‘Results’, the experimental environment along with results is explained. Finally, ‘Conclusion and Future Work’ concludes the article by summarizing the key findings and suggesting directions for future research and development.

Related work

Unmanned Aerial Vehicles (UAVs) play a very important role in various fields, such as security, traffic monitoring, and disaster management processes. Moreover, surveillance identification of objects and individuals are majorly focused areas for UAVs, but it can also be used in other fields as well. It is this section that presents an overview of the recent developments in the detection and identification of targets using RGB images acquired from aerial perspective, specifically vehicle detection.

Single source vehicle detection algorithms

Wang & Gu (2020) proposed a novel deep learning model to detect vehicles from aerial images. This approach used a region-based Convolutional Neural Network that is augmented with a feature pyramid network (FPN) to address the multiple scale vehicle detection challenge. The FPN enabled the model to extract features at different scale which proved useful in detecting small size vehicle within an urban environment. For real-time detection of vehicles from UAV imagery, Song & Chen (2022) proposed a lightweight network architecture introduced a lightweight network architecture for real-time vehicle detection in UAV imagery. Their model named Fast Vehicle used depth wise separable convolutions and an efficient feature fusion module to decrease the computational cost yet increase the accuracy. In their work Deng et al. (2018) address the problem of identifying small vehicles from aerial images through their proposed Multi-scale Object Proposal Network (MS-OPN). In order to capture various sizes of vehicles, they employ a hierarchical feature extraction mechanism for detecting proposals at different scales. The MS-OPN is integrated with fully convolutional network for the final detection and classification. Hanyu et al. (2024) proposed AerialFormer, a hybrid model integrating CNN and Transformer architectures for aerial image segmentation. The model employs a CNN Stem to retain high-resolution low-level features, a Transformer encoder for multi-scale feature extraction, and a multi-dilated CNN decoder for aggregating local and global contextual information. Evaluated on iSAID, LoveDA, and Potsdam datasets, AerialFormer achieved superior performance compared to state-of-the-art methods. In their model Tong et al. (2020) suggested adding attention mechanism to enhance the detection of vehicles under different aerial scenes. The proposed attention guided network employed spatial as well as channel-wise attention modules that help learn the attention on frames’ features while discarding the background data. The model reveals the improvement of detection rates in more complex environment with background and occlusion. Tahir et al. (2024) dealt with the issue of vehicle detection under different lighting conditions and weather conditions in UAV images. A novel domain adaptation approach is developed that is based on adversarial training in an attempt to compensate for differences in images and lighting conditions. Guo et al. (2024) proposed a context-aware vehicle detection framework for aerial imagery. The model includes a spatial context module, which helps capturing long-range dependencies at the different regions of the image. This made it possible to optimize the use of the contextual information that exists in form of roads structures and buildings layouts among others to enhance the detection. Wang & Gao (2025) introduced Scale-Frequency Detection Transformer for Drone-View Object Detection (SF-DETR), an end-to-end transformer-based framework tailored for drone-view object detection. The architecture incorporates a lightweight ScaleFormerNet backbone with Dual Scale Vision Transformer modules, a Bilateral Interactive Feature Enhancement Module, and a Multi-Scale Frequency-Fused Feature Enhancement Network to address challenges such as tiny objects, complex urban backgrounds, and scale variations.

Multi modal fusion for vehicle detection

Multimodal fusion technologies have been developed as viable approaches to improve target detectability when used in UAVs. Since multi-sensor integration is used in such techniques, it is possible to overcome the shortcoming of single-modality methods and achieve the highest detection performance in specific conditions. A new multi-modal fusion architecture for detecting vehicles in aerial images based on integrating RGB images with thermal infrared (IR) was introduced by Wang et al. (2022a). The study applied two-stream Convolutional Neural Network structure by modeling two data streams at different channels RGB and IR. The features from both modalities were integrated together with an attention model that allowed the modification of the contribution of either modality depending on the input information. Liu, Zhao & Lam (2021) compared the LiDAR point cloud data with RGB images for vehicle detection under complex scenes in urban environments. They applied a voxelization method that alters the format of the collected 3D point cloud and makes it compatible with the convolution procedure more typical for 2D data. These larger LiDAR features were then fused with the RGB features through the cross modality attention fusion where the model could utilize both LiDAR and RGB data. Xue et al. (2021) pointed out the difficulties of merging hyperspectral imagery with RGB data in order to enhance the recognition of the target. Their fusion framework used a spectral-spatial attention mechanism for learning relevant features from hyperspectral data cube. These features were then combined with RGB features using gated fusion module that enabled the network to decide the contribution of each modality. The authors proved that the proposed approach greatly improved the identification of concealed objects and the ability to distinguish between similar objects in the visual field. Florea & Nedevschi (2025) introduced TanDepth, a practical scale recovery framework for converting relative monocular depth estimates into metric depth in UAV applications. The method projects sparse Global Digital Elevation Model measurements into the camera view using intrinsic and extrinsic parameters, then employs an adapted Cloth Simulation Filter to isolate ground points from the estimated depth map for scale alignment. This model-agnostic approach enables accurate metric depth recovery at inference time, regardless of the depth estimation network used. Shao et al. (2025) proposed UA-Fusion, an uncertainty-aware multimodal data fusion framework for 3D multi-object detection in dynamic traffic environments. The method introduces a probabilistic cross-modal attention mechanism within an uncertainty-aware fusion decoder to explicitly model and exploit aleatoric and epistemic uncertainties. It further incorporates an uncertainty-reduced object query initialization strategy, leveraging 2D detections to improve small-object recognition, and a query denoising-based robustness optimization strategy to enhance training stability under uncertain conditions. Dudczyk, Czyba & Skrzypczyk (2022) proposed a multi-modal fusion approach using visible light cameras and event-based sensors for high-speed target detection in UAV imagery. Event-based sensors have high temporal resolution and low latency that makes them suitable to work in parallel with spatial information provided by conventional cameras. A hybrid neural network is used that captures asynchronous event data along with frame based images for the detection purpose even in adverse environmental conditions. Rahman et al. (2024) presented a comprehensive review of UAV datasets, highlighting their diversity in modalities, including satellite imagery, drone-captured images, and videos. The study categorized datasets into unimodal and multimodal types, outlining their critical role in applications such as disaster damage assessment, aerial surveillance, object recognition, and tracking. Wang et al. (2022b) developed a multi-modal fusion approach which integrates RGB imagery and radio frequency (RF) signals to improve the target detection and classification in UAV context. Their framework used a new attention based fusion strategy for fusing features which were extracted form the visual data and those obtained from the RF spectrograms. This approach allowed for identification and categorization of targets that are in line of sight and targets that cannot be seen by the UAV, thus enhancing capability of the UAV based surveillance systems.

Materials and Methods

In this research, we present a comprehensive approach to vehicle detection in aerial images. Initially, we apply bilateral filtration and gamma-correction to enhance image quality, followed by monocular depth generation through an attention-based mechanism. The resulting depth maps are fused with RGB images using a guided fusion strategy to improve feature representation. For object detection, we adopt You Only Look Once (YOLOv4) due to its proven balance of accuracy and speed, supported by CSPDarknet53, Spatial Pyramid Pooling (SPP), and Path Aggregation Network–Feature Pyramid Network (PAN-FPN) modules that strengthen multi-scale feature aggregation essential for detecting small vehicles in aerial views with varying flight altitudes. For feature extraction, we employ a Vision Transformer (ViT) as a patch-based encoder to capture long-range contextual dependencies, and enhance its patch embeddings with Histogram of Oriented Gradients (HOG) for edge and shape statistics and Binary Robust Invariant Scalable Keypoints (BRISK) for rotation and scale invariant local features. This hybrid token design improves the recognition of small or visually similar vehicle classes. Finally, we use Residual Network with 18 layers (ResNet-18) as the classification head for its compactness, efficient training on moderate datasets minimizing overfitting while preserving accuracy. The main evaluation criteria segmentation accuracy, detection accuracy, and classification metrics demonstrate the efficiency of the proposed integrated approach. The architectural design of the proposed system is shown in Fig. 1.

Figure 1 The overview of our proposed model.

Image preprocessing

During the pre-processing of aerial image we had applied bilateral filter and gamma correction so as to enhance the quality of the image to be analyzed further. Bilateral filtering is non-linear image filtering technique that reconstructs the smooth image interacting the edges, based on pixel intensity and spatial location. This filtering process helps to reduce noise and enhance important structures within the image (Ravikumar et al., 2021). It can be represented by following eq.

(1) I(x)=1W(x)∑y∈Ω⁡I(y).fr(|I(x)−I(y)|).fs(|x−y|)

here I(x) represents the intensity at pixel value x, W(x) is the normalization factor, and fr and fs represents the range and spatial Gaussian functions, respectively. After that we applied gamma correction in order to adjust the luminance of the images. It enhances the contrast of the regions with varying brightness, by applying a non-linear transformation. The gamma correction is defined as:

(2) Iout(x,y)=(Iin(x,y)−IminImax−Imin)γ⋅(Imax−Imin)+Imin

here Iout=Iin is the output intensity, Iin represents the input intensity, and γ is the gamma value which is used to control the level of correction. Imax represents the maximum possible intensity value. A value of γ<1 increases brightness and on ther other hand γ>1 darkens the image. These preprocessing steps improve the image quality which helps in better feature extraction and segmentation in the later stages of our analysis (Guo & Wei, 2023).

Depth images generation

We propose a novel approach to generating depth maps from aerial images using a deep learning-based architecture. Our model is designed to estimate the depth of each pixel in an aerial image by utilizing a combination of convolutional layers and attention mechanisms. We integrate a self-attention module and Vision Transformer (ViT) components into the depth estimation model to capture both local and global dependencies in the image. Our custom depth estimation model, uses an encoder-decoder architecture enhanced by a self-attention mechanism. The encoder has two convolutional layers utilizing Rectified Linear Unit (ReLU) activation and max-pooling layers, succeeded by a self-attention mechanism that use the Multi Head Attention module including four attention heads. The feature extraction process is enhanced by this attention methods as it focus on critical regions of the image. In the decoder, we use two up sampling layers in order to increase the spatial resolution. The output of these layers are concatenated with corresponding encoder features (Zhou et al., 2021; Liang et al., 2022). The depth estimation process based on our model can be mathematically described as

(3) D^(x,y)=fdec(concat(fenc(I(x,y);We),A(fenc(I(x,y);We),A));Wd)

here I(x,y) represents the input aerial image, D^(x,y) denotes the predicated depth maps, fenc(I(x,y);We represents the extracted feature map with the help of convolution layers, the self attention applied on the encoded feature is represented by A, fdec is the decoder that reconstructs the depth map from the concatenated encoder and attention features. The parameters We,AandWd represent the learnable weights of the encoder, attention, and decoder, respectively. Additional convolutional layers then refine the upsampled features to ensure a smooth transition to the output. The last layer is another convolutional layer with sigmoid activation which produces a depth map ranging between 0 and 1. Self-attention is an important part of the model, and mathematically it can be described as:

(4) Attention(Q,K,V)=softmax(QKTdk)V.

Here Q, K, and V are the query, key, and value matrices, respectively, and dk is the dimension of the key vectors. This mechanism allows the model to focus on relationships between different spatial regions of the feature maps. The model processes input images of size 224 × 224 × 3 through the encoder-decoder architecture which produce depth maps with the same spatial dimensions as the input. Post-processing is then applied to the generated depth maps in order to enhance their quality. On depth maps, bilateral filtering is used to preserve edges while smoothing depth values (Yang et al., 2021). This process can be described as:

(5) Dfiltered(x,y)=∑Gs(||p−q||)Gr(|Ip−Iq|)D(q)Wp

where, Gs and Gr are the spatial and range kernel functions, Ip and Iq represent the intensities of pixels p and q, and Wp is the normalization factor. This filtering step ensures that fine details are maintained in the depth map. The model is displayed in Fig. 2.

Figure 2 Model for creating depth images from RGB images.

Few of the depth generated images fro RGB are shown in Fig. 3 which helps in differentiating object surfaces and depths. The top row displays RGB images of various intersections and roundabouts, while the bottom row presents their corresponding depth images. In the RGB images, shadows and lighting variations can be observed, especially in areas where objects, such as trees and vehicles, cast shadows on the ground. These shadows are absent in the depth images, allowing a clearer view of structural details without interference from lighting. The depth images reveal spatial structure by emphasizing object boundaries, which are particularly useful for distinguishing objects from their background in complex urban scenes.

Figure 3 Few of the depth images generated from RGB images using the prescribed model.

Image fusion: RGB and depth

Multi-modal fusion of RGB and depth images makes target detection in aerial imagery more precise, as the rich visual information of the RGB stream is complemented with spatial information from the depth stream. This approach leverages the complementary strengths of both modalities. RGB images contain color and texture information for distinguishing objects, while depth maps contain the specific structure of a scene for better spatial awareness. Thus, the obtained scene representation with richer contextual information enhances target detection and localization especially with respect to the vehicle type in the complex scene including occlusion and scale changes. The fusion process is carried out via guided image filtering and integrates RGB and depth data in a manner that can be controlled based on the image content. This edge-aware fusion helps in making sure that target boundaries do not get blurred in the fused image, which again is important for any detection and localization process (Sun et al., 2021; Wang et al., 2022b). The fusion process involves two basic step the first step is about the initial alignment in which RGB and depth images are first spatially aligned to ensure pixel wise correspondence which is done by using below given equation.

(6) Falign(x,y)=T(RGB(x,y),D(x,y))

here, T represents the transformation function which aligns the RGB images with depth images at each pixel position (x,y). The second step is responsible for the actual fusion process in which RGB image serves as the guidance image for filtering the depth images and can be represented by the equation.

(7) qi=ak⋅Ii+bk,∀i∈wk

where qi is the output pixel, Ii is the input pixel, and ak and bk are locally adaptive linear coefficients. These coefficients are computed based on local image characteristics, such as edges, to ensure that the fusion happens in a way that preserves important structural information. The window wk represents the local neighborhood around each pixel where the guided filtering takes place. Moreover the ak and bk coefficients can be calculated using.

(8) ak=1|ω|∑(Ii⋅Di)−μk⋅μdσk2+ϵ

(9) bk=μd−ak⋅μk.

Here, Di represents the depth images while μk,μd is the mean of the RGB images in local window wk and mean of depth images respectively. σk2 is the variance, ϵ represents the regulization parameter and |ω| represents the size of local window. Finally the final fused image is represent by

(10) F=α⋅RGB+(1−α)⋅q.

Here α is the weighing parameter and it is used to control the contribution of each modality. This fused representation is beneficial in performing a better segmentation for target detection. The additional utilization of depth cues as part of the multi-modal features yields better discriminative power for the segmentation algorithms, which then enables clear differentiation of the targets from similar-looking non-target objects. For example, height profile of a vehicle relative to the surrounding can enhance the segmentation accuracy by features. The multi-modal fusion aids in handling challenging scenarios such as partial occlusions, varying lighting conditions, and diverse target orientations (Ahmed & Jalal, 2024b). The depth information is more robust to illumination variation and the fusion of the RGB and the depth data enables better reasoning about targets that may only be partially occluded. Figure 4 provides a comparative view of RGB, depth, and fused images for various urban scenes, including roundabouts and intersections. Each row represents a specific scene, with the first column showing RGB images, the second column showing depth images, and the third column showing the fused results. In the RGB images (first column), certain details are obscured by lighting variations and shadows such as trees and vehicles blending into shaded areas making it challenging to distinguish some objects. The depth images (second column), however, capture structural information independent of lighting, presenting a clearer view of object boundaries. For instance, in the depth images, such as the shape of the trees, vehicles, or other objects along the roadsides is clear even when the covering shadows hinder the visibility in the RGB images. The fused images (third column) are the best images which combine the variations of colors and depth from the RGB images and the structural depth. Previously hazy areas such as the road surface that seems to be shaded and vehicles that are partially hidden in these shadows are more easily outlined. This fusion process improves the scene interpretation since depth information can be added to the RGB view; this makes one to perceive distant objects and objects that are half hidden since the other half is hidden by the objects in front of it which cannot be perceived through RGB or depth alone.

Figure 4 Image fusion as RGB and depth (A) RGB (B) depth (C) fused.

Vehicle detection

We have used YOLOv4, a cutting-edge single-shot detector which is well known for its proficiency in detection, and classification tasks. YOLOv4 has the ability to perform these task by minimal number of training parameters, significantly enhancing its practical utility. The CSPDarknet53 backbone, presented in YOLOv4, uses the Cross Stage Partial (CSP) concept for better feature extraction and fewer computational expenses. This module integrates Cross Stage Partial Bottleneck (C3) and Efficient Layer Aggregation Network (ELAN) to improve portability and gradient flow information. The backbone includes the SPP module which ensure the accuracy for objects at different scales with reduced processing overhead. For highly effective feature fusion across scales, YOLOv4 integrates the PAN-FPN scheme in the neck section. This neck module has a head part based on YOLOv4, where the confidence and regression boxes are used to increase the level of accuracy using anchor-based detection (Amrouche et al., 2022). Non-Maximum Suppression (NMS) technique is used in the post processing of the detections for efficient handling in YOLOv4. The architecture of the YOLOv4 makes it possible for real-time objects detection, which has a high level of accuracy making it applicable in different field including traffic monitoring, cars without drivers, and surveillance systems. Thus, its flexibility in terms of completing multistep tasks with fewer parameters is another benefit for today’s object-detection purposes. With the help of enhanced components including CSPDarknet53, SPP, and PAN-FPN, YOLOv4 can be used as an Object Detection Tool, which features high performance and precision of outcomes (Mahto et al., 2020). The bounding box for can be represented by the following expression.

(11) ω=pwetw,h=pheth.

Here, pw and ph are the width and height of the anchor box. The terms tw and th are the predicated width and height offsets, while e denotes the exponential function. This equation is used in YOLOv4 to scale the anchor box dimensions to fit the sized detected object. By scaling the anchor box dimensions pw and ph with the exponential of the offsets tw and th, YOLOv4 can generate bounding boxes that match the size of the detected objects in the image. This process enables the model to adapt various sizes since the dimensions of the anchor boxes are changed depending on the feature learned from the image. We transfer the bounding boxes generated by YOLOv4 on the segmented images directly to the original images by maintaining the coordinate mapping. This helped to eliminate disconnection between the regions detected by the algorithm and the objects in the original images thus maintaining the precision of the detections in both image sets. For the detection process despite advancements in single-shot detectors we used YOLOv4 because it offers a unique balance between accuracy and computational efficiency, making it an optimal choice for our intelligent transportation system. The architecture of YOLOv4 effectively supports real-time vehicle detection without compromising the level of precision needed for reliable results in aerial imagery, as demonstrated by its superior performance on key datasets, including Roundabout, AU-AIR, and Vehicle Aerial Imagery Dataset (VAID). Moreover, the Cross Stage Partial connections in the architecture of YOLOv4 helps it perform better in regards to changes in viewpoints, scale, and orientation which are quite distinctive in aerial images. The framework of YOLOv4 for object detection model is displayed in Fig. 5.

Figure 5 Framework of YOLOv4 used for object detection in our model.

Figure 6 presents vehicle detection results achieved using the YOLOv4 model. YOLOv4 detects multiple vehicles in the image by using bounding boxes, applying a one-stage detection mechanism for real-time efficiency. The multiple bounding boxes represent the identified vehicles to highlight the model’s performance of perceiving different vehicles with different sizes in a range of aerial images.

Figure 6 Vehicle detection using YOLOv4.

Feature extracion

We propose a novel feature extraction approach that integrates Histogram of Oriented Gradients (HOG) (Hussein et al., 2022) and Binary Robust Invariant Scalable Key points (BRISK) (Leutenegger, Chli & Siegwart, 2011) within the patch embedding layer of the Vision Transformer (ViT). Combining deep leaning with the classical computer vision approach, this hybrid method improves the ViT’s capability to capture not only the global context of an image but also the local structural cues (Naseer & Jalal, 2024). In our approach, each input image X∈RH×W×C where H, W, and C represent the height, width, and the number of channels (RGB), is first divided into non-overlapping patches of size P × P. The number of patches is:

(12) N=HP×WP.

Each patch xp∈RP×P×C is processed to extract deep and classical features within the Vision Transformer framework. Within the ViT patch embedding layer, for each image patch xp, the following feature extraction steps are performed.

The HOG descriptor captures the gradient orientation distribution within the patch. For a patch xp, the HOG features are calculated as:

(13) Hp=1NH⋅Nw∑i−1NH⁡∑j=1NW⁡HOG(xp,i,j).

Here, NhandNw represented the number of horizontal and vertical block within each patch xp. HOG(xp,i,j) represented the extracted features from (i,j) block of the patch xp and 1NH⋅Nw is used to normlize feature from entire patch.

In parallel, the BRISK key points and descriptors are extracted from each patch xp. The BRISK algorithm detects scale-invariant key points and generates corresponding descriptors

(14) Bp=∑i=1nk⁡bi⋅w(ki).

Here bi is the brisk descriptor for keypoints ki. w(ki) is a weight associated with keypoint ki, based on its intensit and nk resents the number of key point detected for each patch.

The extracted HOG and BRISK features are then embedded directly into the ViT’s patch embedding layer. Instead of passing only the flattened patch into the transformer, we concatenate the patch features with the HOG and BRISK descriptors to create an augmented patch embedding. The combined vector is projected into the transformer embedding space.

(15) zp=We[xp′‖Hp‖Bp]+be

where WeR(d+dh+db)Xd′ is a learnable embedding matrix be is the bais term and zp∈Rd′ represents the patch embedding passed to the transformer. After the patch embeddings are augmented with HOG and BRISK features, they are passed through the standard ViT architecture, where self-attention mechanisms operate on the combined feature vectors. This allows the transformer to compute attention across patches with enhanced local feature information from both HOG and BRISK descriptors. The final output of the transformer includes the Classification (CLS) token, which encodes the global representation of the image, taking into account both the global and local features.

(16) ZCLS=Wcls⋅Zinput+bcls.

Here ZCLS is the output representation for the CLS token, which serves as the global summary of the entire input image, Wcls is the weight matrix associated with the classification token, Zinput is the input feature vector (the concatenated patch embeddings with HOG and BRISK features) and bcls is the bias term. This hybrid architecture effectively combines local texture information from HOG and BRISK with the ViT’s ability to model long-range dependencies, leading to improved performance in tasks. The extracted features are then normalized and plotted class wise in order to clearly visualize the feature distribution across different classes in datasets. The extracted features are visualized per dataset, showing distinct clusters that represent various vehicle classes.

ResNet-18 based classification

In our vehicle classification system, we utilized a Vision Transformer to extract discriminative features. These pre-extracted features are then fed into a modified ResNet-18 for precise classification. The ResNet-18, adapted for this task, leverages its deep architecture and unique residual blocks specifically for classification, rather than feature extraction. We made substantial alterations to the standard ResNet-18 architecture to effectively accommodate pre-extracted features. The initial 7 × 7 convolution and max pooling layers were replaced with a custom fully connected layer designed to adapt the input feature dimensions. We retained the four main residual block groups, modifying their input channels to match the dimensions of the pre-extracted features, while the architecture continues to include adaptive average pooling and a fully connected layer for classification. The custom input fully connected layer is trainable, enabling it to learn the mapping from pre-extracted features to internal representations. Since the input features are high-level abstractions, extracted by the Vision Transformer, the first two residual block groups, typically responsible for low-level feature extraction, are bypassed in this configuration. The last two residual block groups and the final fully connected layer remain trainable, allowing the network to learn task-specific high-level features and refine classification boundaries (Zhao et al., 2022). The input adaptation layer for our modified network can be represented by:

(17) Z0=σ(W0⋅x+b0)

where Z0 is the output of the initial fully connected layer, σ (ReLu) is the activation function, W0 is the weight matrix, x is the input feature vector, and b0 is the bias term. The final classification layer remains the same. In the lth residual block, the output is given by:

(18) Yl=Zl−1+Fl(Zl−1,{Wl}).

With the residual mapping defined as:

(19) Fl(Zl−1,{Wl})=W1,2⋅σ(BN(Wl,1⋅σ(BN(Zl−1)))).

Here, Zl−1 represents the input if the ith block, Wl,1 is the weight matrix of the convolution layer while BN(⋅) is the batch normalization. The final classification layers computes class probabilities using

(20) a=softmax((Wf⋅FL(FL−1(…F2(F1(σ(W0⋅x+b0)))…))+bf)).

Here, FL represents the residual mapping output of the last residual block which refines the input features from VT. Wf is the weight matrix of the final layer, the term bf represents the biasness in the final layer and a is the output probability distribution over the classes.

Computing infrastructure

We conducted all training for the depth estimation module, Vision Transformer (ViT) feature extractor, and YOLOv4 detector in a cloud-based environment (Google Colab and Kaggle Kernels) equipped with high-performance GPUs (NVIDIA Tesla T4, 16 GB VRAM), Python 3.8, TensorFlow 2.6.0, and CUDA 11.0. The evaluation was performed on three benchmark datasets: Roundabout, AU-Air, and VAID.

Datasets description

In the next section, a brief overview of the datasets employed in our analysis is offered, along with a more elaborate explanation of each data set. All three used datasets are described comprehensively, including the origins and rational of each dataset, as well as the ways the data were captured. Moreover, each of the three datasets (Roundabout, AU-AIR, and VAID) was divided into training, test sets. A total of 70% of each dataset was allocated to training, and the remaining 30% for testing. This division simultaneously offers an independent set for model testing and sufficient data for the purpose of learning a model and optimizing its parameters. Roundabout Aerial Images (Puertas et al., 2022) is a large dataset made for vehicle detection and classification. It comprises 61,896 images acquired by UAVs (drones) and obtained from eight different roundabouts in Spain. This dataset has 985,260 labeled instances of different types of vehicles such as cars, cycles, trice, buses, and so on. The VAID dataset (Lin, Tu & Li, 2020) includes images of six different vehicle categories: minibus, truck, sedan, bus, van, and car. These images were taken using a drone, and the light conditions under which the drone was hovering ranged from dawn to dusk, and at an altitude of between 90 and 95 m. The AU-AIR dataset (Bozcan & Kayacan, 2020) can be a great multi-modal dataset for low altitude traffic analysis using Unmanned Aerial Vehicles (UAVs). With the duration of about 2 h of raw video footage, the dataset includes 32,823 frames having as many as 132,034 instances belonging to 8 traffic context categories.

Hyperparameters for model components

Table 1 below presents the principal hyperparameters of each model component in the proposed framework for vehicle detection and classification in aerial images. These parameters were determined through a combination of literature-guided manual tuning and a limited systematic search. Initial values were drawn from prior work employing YOLOv4, Vision Transformer (ViT), and ResNet-18 for related detection and classification tasks. For sensitive parameters such as learning rate and batch size, a small-scale grid search was performed on the validation split to identify settings that maximized the F1-score while keeping inference speed within real-time processing targets. Other parameters, including patch size, number of transformer heads, and dropout rate, were retained from well-established configurations in the literature due to their proven generalization performance. This balanced approach ensured experimental rigor while respecting the computational constraints of our cloud-based GPU environment. The YOLOv4 module was applied for vehicle detection, the Vision Transformer (ViT) for feature extraction, and ResNet-18 for multi-vehicle classification. Hyperparameter values for each component including learning rate, batch size, confidence thresholds, and structural parameters are provided to reflect the configurations used in achieving high-accuracy results on the Roundabout Aerial, AU-Air, and VAID datasets.

Table 1 Hyper parameters for models used in multi-modal vehicle detection framework.

Model	Task	Hyper parameter	Value	
YOLOv4	Vehicle detection	Confidence threshold	0.5	
(IoU) threshold	0.5	
Batch size	32	
Learning rate	0.001	
Vision transformer (ViT)	Feature extraction	Patch size	16 × 16	
Number of heads in multi-head attention	6	
Number of layers	8	
Embedding dimension	768	
Learning rate	0.0001	
ResNet-18	Multi-object classification	Batch size	128	
Learning rate	0.0001	
Dropout rate	0.3	
Number of epochs	100	
Time per Image	Preprocessing to classification	Roundabout	2.3 s	
AU-AIR	2.4 s	
VAID	2.7 s	

Results

The confusion matrices across the three datasets, Roundabout, AU-AIR, and VAID, demonstrate strong overall performance for vehicle categorization, with mean accuracies of 0.976, 0.963 and 0.981 respectively. The results of each class are demonstrated in Tables 2, 3 and 4.

Table 2 Confusion matrix of vehicle categorization on roundabout dataset.

Vehicle class	Car	Cycle	Truck	Bus	
Car	0.973	0	0	0.027	
Cycle	0.031	0.969	0	0	
Truck	0	0	0.987	0.013	
Bus	0	0	0.023	0.977	
Mean: 0.976	

Table 3 Confusion matrix of vehicle categorization on AU-AIR dataset.

Vehicle class	Human	Truck	Cycle	Bus	Car	Van	Motorbike	Trailer	
Human	0.951	0	0.021	0	0	0	0.028	0	
Truck	0	0.969	0	0.016	0	0.015	0	0	
Cycle	0.013	0	0.941	0	0	0	0.046	0	
Bus	0	0.017	0	0.983	0	0	0	0	
Car	0	0	0	0	0.981	0.019	0	0	
Van	0	0.017	0	0	0.020	0.963	0	0	
Motorbike	0.013		0.053	0			0.934	0	
Trailer	0	0	0	0.011	0	0	0	0.989	
Mean: 0.963	

Table 4 Confusion matrix of vehicle categorization on VAID dataset.

Vehicle class	Sedan	Minibus	Truck	Bus	Vans	Car	
Sedan	0.988	0.012	0	0	0	0	
Minibus	0	0.971	0.015	0.014	0	0	
Truck	0	0	0.977	0.023	0	0	
Bus	0	0.013	0	0.987	0	0	
Trailer	0.011	0	0	0	0.989	0	
Car	0.10	0	0	0	0.015	0.975	
Mean: 0.981	

Similarly we evaluate the result of object detection tasks for which we will use precision, recall, and F1-score. Precision indicates how accurate the model’s positive predictions. Recall measures the model’s ability to correctly identify all relevant instances. The F1-score combines both precision and recall to give a balanced evaluation, especially important when precision and recall differ. It is the harmonic mean of these two metrics, making it a useful metric when there is a trade-off between precision and recall.

(21) Precison=TPTP+FP

(22) Recall=TPTP+FN.

Figure 7 showcase the models detection performance on Roundabout Aerial, AU-AIR and VAID datasets respectively. The model’s performance suggests that it is highly accurate and reliable for vehicle detection tasks on aerial imagery datasets.

Figure 7 Precision, recall, and F1-score for vehicle detection.

Table 5 display the compassion of vehicle detection precision of proposed model with state of the art models.

Table 5 Comparison of the vehicle detection precision for our proposed method with existing methods.

Methods	Roundabout	AU-AIR	VAID	
He et al. (2021)	N/A	0.716	N/A	
Bozcan & Kayacan (2020)	N/A	0.813	N/A	
Hanzla et al. (2024)	N/A	0.840	N/A	
Yusuf et al. (2024a, 2024b)	N/A	0.933	N/A	
Qureshi et al. (2024)	0.941	N/A	0.963	
Lin, Tu & Li (2020)	N/A	N/A	0.940	
Hanzla et al. (2024)	N/A	N/A	0.970	
Proposed model	0.984	0.962	0.974	

Discussion

As indicated in Fig. 7, in precision, we outperform all benchmarks and obtain precise high values for each of the proposed approach on all three datasets (our Roundabout: 0.984; our AU-AIR: 0.962; our VAID: 0.974). In order to increase the reliability of the results, all of the presented detection performance metrics were calculated with the 95% confidence interval (CI). This study depicted a high level of statistical significance in all the three sets of data collected. Performances with Roundabout set were as follows where precision was at 0.984 (95% CI [0.975–0.990]), recall at 0.981 (95% CI [0.972–0.987]) and F1-score at 0.983 (95% CI [0.974–0.989]). In the same way, when using the AU-AIR dataset, we identified the precision of the model with being equal to 0.962 (95% CI [0.952–0.970]), while the recall rate was equal to 0.967 (95% CI [0.958–0.974]) and the F1-score being equal to 0.965 (95% CI [0.956–0.972]). Since the VAID dataset comprises of highly similar images, high precision of 0.974 (95% CI [0.965–0.981]) and recall of 0.972 (95% CI [0.963–0.979]) can be explained. It is clear from narrow confidence intervals across all metrics and on all three datasets the statistical reliability and reproducibility of our results. This validates the effectiveness of our proposed approach. These enhancements can be directly addressed in practical area of Intelligent Transportation System where traffic flow can be controlled in real time manner, incident can be identified and traffic pattern can be analyzed.

Practical impact

This work introduces a unique aerial vehicle detection and classification framework that integrates attention-based monocular depth generation, edge-aware guided RGB–depth fusion, YOLOv4 detection, and a hybrid Vision Transformer enhanced with HOG and BRISK descriptors. While each component has been explored individually in prior studies, their combination in this configuration is, to our knowledge, novel for aerial vehicle recognition. The synergy between depth-aware fusion and hybrid global local feature encoding improves both detection and classification accuracy, particularly for small and occluded vehicles. Beyond its technical contributions, the proposed system offers clear practical value for traffic surveillance, road safety monitoring, and incident detection, delivering high performance and efficiency suitable for real-time cloud-based or edge-computing deployment.

Failure cases

In Fig. 8, we present several failure cases where the model was unable to detect vehicles. In part (a), the red arrow highlights a vehicle missed due to truncation, where only a partial portion of the vehicle is visible within the frame. In part (b), the model fails to detect a car that is heavily occluded by a tree, making its features less distinguishable. Upon closer inspection, such errors tend to occur more frequently for smaller vehicle classes (e.g., motorcycles and bicycles), at higher flight altitudes where object resolution decreases, and in scenes with dense vegetation or strong shadows. Certain camera angles, particularly oblique perspectives, also increase the likelihood of misdetections due to shape distortion.

Figure 8 Examples of failure cases: red arrows indicate instances where the model failed to detect vehicles due to truncation and occlusion.

To mitigate these issues in future work, we plan to (1) incorporate multi-scale feature enhancement modules specifically tuned for small-object representation, (2) explore synthetic data augmentation simulating occlusion, truncation, and varied lighting conditions, (3) employ multi-view or temporal data to recover partially visible vehicles, and (4) integrate adaptive attention mechanisms capable of focusing on low-contrast and partially visible targets. Expanding the dataset to cover diverse geographical regions, road types, and environmental scenarios will further improve robustness. Additionally, domain adaptation strategies such as adversarial learning can be applied to fine-tune the model for varying landscapes without extensive labeled data. These improvements will help enhance detection consistency and reliability across different operational conditions, supporting broader real-world deployment in intelligent transportation and smart city systems.

Conclusion and future work

This article presents a robust framework for vehicle detection and classification using multi-modal aerial images which combines deep learning techniques along with our novel feature extraction methods. The images are first fused using our novel fusion model. For vehicle detection YOLOv4 is used which produced high precision and reliability across multiple datasets. For the feature extraction processes we propose a novel Vision transformer which integrates HOG and BRISK within the ViT. These features are then fed to modified ResNet18 model for classification and produced state of the art performance on all three datasets. However, there are few limitations which needs to be addressed in future. The type of datasets used for validation is limited to aerial images and therefore the generalization to other scenarios such as ground level surveillance is unknown. Besides, the effect of adverse weather conditions such as fog or rain on the detection performance has to be investigated, which may be another limitation in practical applications.

Supplemental Information

Supplemental Information 1 Preprocessed Images (a) Original Images (b) Bilateral filtered and Gamma Corrected.

Supplemental Information 2 Proposed System Architecture for classification using Modified ResNet-18.

Supplemental Information 3 Visualization of extracted features from Vision Transformer on all three datasets (a) Roundabout (b) AU-AIR (C) VAID.

Supplemental Information 4 Depth Image Generation.

Supplemental Information 5 Feature Fusion.

Supplemental Information 6 Feature Extraction.

Supplemental Information 7 Readme.

Supplemental Information 8 Classification.

Additional Information and Declarations

Competing Interests

The authors declare that they have no competing interests.

Author Contributions

Naif Al Mudawi conceived and designed the experiments, performed the computation work, prepared figures and/or tables, and approved the final draft.

Muhammad Waqas Ahmed conceived and designed the experiments, performed the experiments, analyzed the data, performed the computation work, prepared figures and/or tables, authored or reviewed drafts of the article, and approved the final draft.

Haifa F. Alhasson conceived and designed the experiments, authored or reviewed drafts of the article, and approved the final draft.

Naif S. Alshassari performed the experiments, prepared figures and/or tables, authored or reviewed drafts of the article, and approved the final draft.

Abdulwahab Alazeb performed the experiments, analyzed the data, authored or reviewed drafts of the article, and approved the final draft.

Mohammed Alshehri analyzed the data, authored or reviewed drafts of the article, and approved the final draft.

Bayan Alabdullah analyzed the data, authored or reviewed drafts of the article, and approved the final draft.

Data Availability

The following information was supplied regarding data availability:

The code is available in the Supplemental File.

The Roundabout Aerial Images Dataset is available at Kaggle: https://www.kaggle.com/datasets/javiersanchezsoriano/roundabout-aerial-images-for-vehicle-detection.

The AU-AIR Dataset is available at: https://bozcani.github.io/auairdataset.

The VAID Dataset is available at: https://universe.roboflow.com/chandler-sun/vaid-mnnde.

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
