# Peer review of "Multimodal image fusion for enhanced vehicle identification in intelligent transport"

_PeerJ Computer Science, doi:10.7717/peerj-cs.3270_

## Round 0.1 · original submission · Major Revisions

Please follow all requests and address all criticisms thoroughly.

**Language Note:** The review process has identified that the English language must be improved. PeerJ can provide language editing services - please contact us at [email protected] for pricing (be sure to provide your manuscript number and title). Alternatively, you should make your own arrangements to improve the language quality and provide details in your response letter. – PeerJ Staff

·

Basic reporting

- It is recommended that all techniques, acronyms, and methods used be defined at first use to facilitate reading, especially for readers who are not completely familiar with all terms (e.g., briefly explain acronyms such as HOG, BRISK, ViT). As an example, the first occurrence of HOG is on line 40, but is nevertheless developed on line 106.

- Although the references are adequate and relevant, the state of the art could be strengthened by including some very recent (last 1-2 years) citations on multimodal sensing or ViT applications in airborne vision, to place the work even better within the current context of the field.

Experimental design

- The paper uses a combination of advanced techniques (YOLOv4, Vision Transformer with HOG and BRISK, ResNet-18). It is recommended to include a brief discussion justifying why these specific architectures were selected over other possible alternatives.

- It would also be valuable to indicate whether other architectures or approaches were evaluated, and what the decision criteria were for opting for the chosen models.

- Table 1 provides the values of the key hyperparameters. It would be useful to briefly explain whether any systematic search for hyperparameters was performed (e.g., grid search, random search, Bayesian optimization) or whether the values were established manually based on previous work. This would strengthen the perception of rigor in the experimental design.

Validity of the findings

- The paper shows that their approach is better than other methods for detecting and classifying vehicles in aerial images, but it doesn't talk about how new this approach is or what the impact might be. It is suggested to add a section or subsection within the Discussion or Conclusion that clearly highlights how this approach is new (e.g., combining depth map generation with attention, guided filtering, YOLOv4, and HOG/BRISK in ViT) and discusses how it can be used in practice, for example, for traffic surveillance or road safety.

Additional comments

- Although the cases where the model fails (e.g., truncation and occlusion of vehicles) are identified, this section could be enriched by further characterizing these errors (are they more frequent in certain classes of vehicles, flight heights, camera angles, light conditions, etc.) and proposing specific ideas to mitigate them in future works.

- It would be convenient to review the uniformity in the format of the figures (same decimal notation, same color coding in similar graphs, etc.) to improve the visual presentation of the article.

Reviewer 2 ·

Basic reporting

• Clear English is used, but minor grammatical errors exist (such as "In there work" to "In their work," ).
• Introduction adequately motivates the study but could better highlight gaps addressed by the proposed fusion method.
• Structure aligns with PeerJ standards, though the "Related Work" section lacks depth in multi-modal fusion.

Experimental design

• Methods are described with sufficient detail for replication, including preprocessing (bilateral filter, gamma correction), depth generation (encoder-decoder + attention), fusion (guided filtering), and feature extraction (ViT + HOG/BRISK).

• Hyperparameters are well-documented (Table 1), and datasets are publicly accessible.

• Evaluation metrics (precision, recall, F1-score, confusion matrices) are appropriate.

• Computational infrastructure (Intel i3, 4GB RAM) is inadequate for training ViT/YOLOv4. Clarify if GPUs were used, as this impacts reproducibility and real-time claims.

• Depth map generation lacks training details (e.g., loss function, dataset). Specify if depth maps were synthetically generated or used as ground truth.

Validity of the findings

General Comments

• Fig. 1: Simplify the workflow diagram to clarify data flow (e.g., depth generation → fusion → detection).

• Table 5: Include missing comparisons (e.g., VAID for He et al. 2021) and standardize reference formatting ("Lin et al." → "Lin et al.").

• Preprocessing: Specify parameters for bilateral filtering (σ) and gamma correction (γ).

• Feature Extraction: Justify HOG/BRISK integration in ViT (e.g., how they enhance local features).

• Class Imbalance: Report per-class precision/recall to address misclassifications (e.g., AU-AIR "Motorbike" to "Cycle").

• Grammar: Fix tense inconsistencies (e.g., "We transferred" to "We transfer," ).

References: Ensure all citations match the reference list (e.g., "Ahmed et al. 2024" in text vs. "Ahmed and Jalal 2024" in Refs).

---

## Round 0.2 · accepted · Accept

Congratulations on your valuable contribution to our journal.

·

Basic reporting

no comment

Experimental design

no comment

Validity of the findings

no comment

Additional comments

I am satisfied with the new version of this manuscript and my advice is this paper can be accepted.

Reviewer 2 ·

Basic reporting

Address all the basic concepts that are necessary for that article.

Experimental design

The experimental design is now clear and follows the aim of the journal.

Validity of the findings

This articles have suitable novelty that is necessary for publication and dominance.